# Transcriptome-Based Identification of a Functional *Fasciola hepatica* Carboxylesterase B

**DOI:** 10.3390/pathogens10111454

**Published:** 2021-11-10

**Authors:** Yaretzi J. Pedroza-Gómez, Raquel Cossio-Bayugar, Hugo Aguilar-Díaz, Silvana Scarcella, Enrique Reynaud, María del Rayo Sanchez-Carbente, Verónica Narváez-Padilla, Estefan Miranda-Miranda

**Affiliations:** 1Centro Nacional de Investigaciones Disciplinarias en Salud Animal e Inocuidad, Instituto Nacional de Investigaciones Forestales Agrícolas y Pecuarias (INIFAP), Boulevard Cuauhnahuac No. 8534, Jiutepec 62550, Morelos, Mexico; yaretzijpg@gmail.com (Y.J.P.-G.); cossio.raquel@inifap.gob.mx (R.C.-B.); aguilar.hugo@inifap.gob.mx (H.A.-D.); 2Facultad de Ciencias Veterinarias, Campus Universitario, Universidad Nacional del Centro de la Provincia de Buenos Aires, Tandil 7000, Argentina; silvanas@vet.unicen.edu.ar; 3Departamento de Genética del Desarrollo y Fisiología Molecular, Instituto de Biotecnología, Universidad Nacional Autónoma de México, Avenida Universidad, 2001, Apartado Postal 510–3, Cuernavaca 62210, Morelos, Mexico; enrique@ibt.unam.mx; 4Centro de Investigación en Biotecnología de la Universidad Autónoma del Estado de Morelos, Av. Universidad 1001, Cuernavaca 62209, Morelos, Mexico; maria.sanchez@uaem.mx; 5Centro de Investigación en Dinámica Celular de la Universidad Autónoma del Estado de Morelos, Cuernavaca Morelos México, Av. Universidad 1001, Chamilpa, Cuernavaca 62210, Morelos, Mexico; vaezgm@gmail.com

**Keywords:** Fasciolosis, bioinformatics, transcriptome, carboxylesterase, zymogram

## Abstract

Bioinformatics analysis of the complete transcriptome of *Fasciola hepatica,* identified a total of ten putative carboxylesterase transcripts, including a 3146 bp mRNA transcript coding a 2205 bp open reading frame that translates into a protein of 735 amino acids, resulting in a predicted protein mass of 83.5 kDa and a putative carboxylesterase B enzyme. The gene coding for this enzyme was found in two reported *F. hepatica* complete genomes stretching 23,230 bp, containing two exons of 1282 and 1864 bp, respectively, as well as a 20,084 bp intron between the exons. The enzymatic activity was experimentally assayed on *F. hepatica* protein extracts by SDS-PAGE zymograms using synthetic chromogenic substrates, confirming both the theoretical molecular weight and carboxylesterase enzymatic activity. Further bioinformatics predicted that this enzyme is an integral component of the cellular membrane that should be active as a 167 kDa homodimer complex and polyacrylamide gel electrophoresis (PAGE) zymograms experiments confirmed the analysis. Additional bioinformatics analysis showed that DNA sequences that code for this particular enzyme are highly conserved in other parasitic trematodes, although they are labeled hypothetical proteins.

## 1. Introduction

The liver fluke *Fasciola hepatica* is a worldwide zoonotic parasitic trematode affecting a vast range of animal hosts, including humans. It is the causative agent of fascioliasis, a parasitic disease highly prevalent in the livestock industry and capable of parasitizing millions of people worldwide [1]. Digenetic trematodes such as *F. hepatica* require a gastropod mollusk as an intermediate host to complete a complex biological cycle; for this reason, fascioliasis distribution is closely related to the intermediate host distribution worldwide [2].

The prevention of fascioliasis is achieved by the use of anthelmintics on the livestock, a procedure that sets the conditions for anthelmintic resistance selection in the liver fluke populations [3,4,5]. *F. hepatica* is an example of the ever-growing resistance to the effects of antiparasitic compounds such as benzimidazole-derived anthelmintics. Previous research on the subject suggests that several xenobiotic metabolizing enzymes (XMEs) may be associated with the resistance phenomenon [6]. Xenobiotic metabolism is responsible for the defense of most organisms against poisonous effects of naturally occurring toxins and the enzymes responsible for this type of metabolism are the focus of scientific scrutiny because they mediate the ever-increasing tolerance and resistance of parasites against antiparasitic agents that plague livestock and humans [7].

Carboxylesterases are a family of XMEs found in all types of organisms [8]; these enzymes hydrolyze xenobiotic compounds bearing chemical functional groups such as carboxylic acid ester, amide and thioester into products containing alcohols and organic acids that then become easily disposable by other supplementary metabolic pathways [9]. 

Carboxylesterases are implicated in diverse biological functions, in addition to their role in the detoxification of xenobiotics, most notably juvenile hormone metabolism, which regulates the development of arthropods and other invertebrates [10]. 

The expression and enzymatic specific activity of carboxylesterases have been demonstrated to play a significant role in the resistance mechanism of a variety of parasites against antiparasitic compounds, such as the *Plasmodium falciparum* response against antimalarial compounds [11,12], insecticide resistance in the malaria vector *Anopheles funestus* [13], malathion resistance in the triatomid vector of American trypanosomiasis *Rhodnius prolixus* [14], and organophosphate resistance in the oriental fruit fly *Bactrocella dorsalis* [15]. Scientific evidence on carboxylesterases associated with insecticide resistance made this XME a reliable molecular marker for resistance screening in populations of the encephalitis vector, the Florida SLE mosquito *Culex nigripalpus* [16]. 

Although there is no previous study of *F. hepatica* carboxylesterase enzymatic activity that may demonstrate a probable linkage to anthelmintic resistance, it has been reported that carboxylesterase enzymatic specific activity in *F. hepatica* is increased in adult parasites recovered from anthelmintic-treated sheep hosts [17], during this study, some basic biochemical properties of *F. hepatica* carboxylesterase, such as the mass and catalytic activity on synthetic chromogenic substrates, were identified for the first time using zymogram assays on cytosolic protein extracts [17]. However, the lack of information on the DNA sequence of this particular gene represents an obstacle to understanding the complementary molecular properties of this enzyme or its probable role in the metabolism of anthelmintics [18]. Consequently, it is the objective of this study is to identify *F. hepatica* carboxylesterase mRNA transcript sequences using high-throughput transcriptome sequencing of total mRNA of the parasite. Once located, a comprehensive bioinformatics analysis of the gene DNA sequence features and its mRNA transcript as well as the properties of the predicted expressed protein is performed and, finally, the predicted proteomic results are compared to experimentally obtained biochemical properties from the actual enzyme found in parasite protein extracts.

## 2. Results

### 2.1. Carboxylesterase Enzymatic Assays and Zymograms

(1) Carboxylesterase enzymatic activity on synthetic chromogenic substrate, demonstrated the presence of high levels of enzymatic specific activity in three different parasite extracts, showing 3.1, 8.3 and 4.7 μmol/mg/min hydrolyzed substrate for the crude extract, membrane fraction and soluble fraction, respectively (Figure 1). 

(2) The SDS–PAGE zymograms showed two isozymes exhibiting carboxylesterase-specific activity as bands of 85 and 170 kDa in the extract’s cytosolic supernatant, whereas only a 170 kDa band was found in the membrane fraction of the protein extract; these data and zymogram are represented in Figure 1. 

### 2.2. F. hepatica Transcriptome

(1) The *F. hepatica* transcriptome database obtained 15530 transripts made public at GenBank BioProject PRJNA679050, BioSample SAMN16822858, RNAseq data SRR13076124 available on-line at https://trace.ncbi.nlm.nih.gov/Traces/sra/?run=SRR13076124. (accessed on 8 November 2021)

(2) The transcriptome sequencing data identified ten transcripts labeled esterase by Gene Ontology/Orthology, and two additional sequences tagged as esterase were not represented in our transcriptome but were found in the *F. hepatica* complete genome datasets available at the GenBank and WormBase-ParaSite databases and described at Table 1. 

(3) The differential expression levels in fragments per million kilobases (FPMKs) from every esterase in our transcriptome determined during the sequencing procedure, were annotated in Table 1 and graphically represented in Figure 2.

### 2.3. Bioinformatics Analysis

(1) All transcripts sequences identified as esterase by the gene ontology enrichment analysis were compared by differential expression to the constitutive gene Beta-Tubulin(14.45 FPMKs) as well as the mRNA size required to translate into an 85 kDa protein or a homodimer complex of 170 kDa proteins determined by SDS–PAGE zymograms. This mRNA, reported as cDNA in the sequencing output, was a 3146 bp transcript containing an open reading frame of 735 codons that was translated into a protein of 735 amino acids by the NCBI Orf finder algorithm (https://www.ncbi.nlm.nih.gov/orffinder/) (accessed on 8 November 2021) (Table 1).

(2) The hypothetical protein coded by transcript 473 was processed for Gene Ontology Enrichment analysis online at http://geneontology.org/docs/go-enrichment-analysis/ accessed on 8 November 2021 (GO:0016021 number), and this enzyme was described as an integral component of the membrane with a probable role of this type of enzyme in a drug metabolic pathway. Additionally, a K03927 KEGG number was obtained at www.genome.jp/dbget-bin/ accessed on 8 November 2021, which describes this enzyme as a carboxylesterase EC:3.1.1.1, 3.1.1.84 and 3.1.1.56, and the results are summarized in Table 1. 

(3) Further bioinformatics results were obtained when the amino acid sequence was analyzed online through the InterPro algorithm (www.ebi.ac.uk/interpro/ accessed on 8 November 2021), which found that the protein exhibited several structural and functional domains, such as a noncytoplasmic carboxylesterase domain at amino acids 97–681, a transmembrane helix at amino acids 73–93 and a cytoplasmic membrane at amino acids 1–73 (Figure 3). Additionally, by comparison to similar enzymes, InterPro predicts a homodimer configuration when attached to membranes (Figure 3). 

(4) Additional online analysis of the amino acid sequence at www.Bioinformatics.org accessed on 8 November 2021 showed that the predicted size of this protein is 83.5 kDa when in a monomeric configuration and 167 kDa when a dimer complex attached to the membrane is formed (Figure 2).

### 2.4. Phylogenetic Analysis

(1) The resulting phylogenetic tree obtained by running pBlast on transcript 473 clearly identifies two major clades with a perfect distinction between cestodes and trematodes; notably the *Fasciola* genus within the trematode clade was set apart in a distinctive sub-clade of its own, where transcript 473 carboxylesterase B, GenBank entry MT843326 can be located with a 99.7% identity with the same protein reported previously and independently at the GenBank, but labeled as hypothetical Genbank entry D915_000180. Also, within the *Fasciola* genus subclade is the equivalent hypothetical protein in *F. gigantica* with a 98% identity. An Esterase F4 of the lepidopteron *Galleria mellonela* was included as an out-group for enhanced accuracy of the phylogeny analysis (Figure 4).

## 3. Discussion

The carboxylesterase of *Fasciola hepatica* analyzed in our study was first reported during a previous zymogram study on cytosolic protein extracts from liver flukes obtained from experimentally parasitized sheep. It was described as a 170 kDa enzyme exhibiting high levels of catalytic activity on synthetic substrates [17]. During this study, we corroborated the presence of high levels of carboxylesterase enzymatic specific activity at 170 kDa in both protein extracts analyzed (Figure 1), as well as an 85 kDa carboxylesterase enzyme not previously reported. It is our interpretation that both enzymes found in the zymogram analysis are actually the same protein showing a monomeric molecular mass of 85 kDa and a homodimeric form of 170 kDa, with both enzymes exhibiting carboxylesterase activity on synthetic chromogenic carboxylester substrates, as demonstrated by zymographic analysis (Figure 1). A more profound interpretation of our zymogram analysis suggests that the 170 kDa carboxylesterase is found in the membrane-bound fraction of proteins, suggesting that this enzyme is a membrane-bound enzyme usually found as a homodimer of 170 kDa, a property common to carboxylesterases in other species [7,20] (Figure 1 and Figure 3).

(1) Since we obtained total mRNA destined for high-throughput transcriptome sequencing from this sample of parasites exhibiting high levels of carboxylesterase-specific enzymatic activity, we assumed that the expression levels of carboxylesterase 85 and 170 kDa isozymes would show a high level of expression with a high value in FMPKs when compared to other carboxylesterases within the same transcriptome. For this purpose, we used the biochemical information obtained from the zymogram interpretation to locate the mRNA sequence of the gene in our transcriptome that could code for a transcript labeled as carboxylesterase that may show two different isoforms consistent with the protein masses and predicted enzymatic specific activity found by SDS–PAGE zymograms.

Among the ten transcripts found in the transcriptome labeled as esterase, three showed a differential expression level close to the constitutive gene Beta-Tubulin transcript 5488, showing an expression level of 19.1 FPMKs in our transcriptome (Figure 2); this transcript was used as a high expression level comparative reference within our transcriptome (Figure 2). Transcript 473 exhibited an expression level of 14.45 FPMK and predicted carboxylesterase activity by gene ontology; the closest transcripts were 4519, showing an expression of 4.11 FPMK, and transcripts 247 and 4520 with 2.6 and 2.1 FPMK, respectively. All three hypothetical proteins showed a predicted enzymatic function as cholinesterase according to the Gene Ontology analysis (Table 1) (Figure 2). All remaining transcripts labeled esterase were discarded as possible candidates due to a low level of expression and/or a predicted incompatible mRNA size or incompatible protein mass when translated.

(2) Although transcripts 247, 4519 and 4520 were considered acceptable expression candidates, their predicted role as cholinesterase makes them incompatible with the enzymatic experimental conditions used during the zymogram assays; nevertheless, the transcripts were further assessed for the predicted number of amino acids when translated as well as the total mass of the expected protein; however, none of them originated an 85 kDa mass protein by the Protein Molecular Weight Calculator algorithm. Only transcript 473 was predicted to translate into a 735 amino acid protein that was used for prediction of protein mass at Bioinformatics.org, obtaining an 83.5 kDa protein for a monomer and 167 kDa for a dimeric complex, almost perfectly matching our experimental calculations by zymograms. The transcript 473 sequence was submitted to NCBI pBlast analysis, and 99.7% identity was found to mRNA 2822 of the *Fasciola hepatica* complete reference genome uploaded to GenBank entry No. LN627097.1, which indicates that this carboxylesterase transcript, although described as a hypothetical protein, has been available for some time. In an additional pBlast against other *Fasciola hepatica* complete genomes, GenBank entries CDMT01004086.1 contig 4086 and OMOY1058219.1 contig 58221 exhibit complete identity in two different regions of both genomes separated in both cases by a DNA space of approximately 20 kbp, these data are consistent with the presence of two exons of 1282 and 1864 bp separated by an unusually large intron of 20 kbp (Figure 3).

(3) The InterPro and Phyre 2 algorithms found a noncytoplasmic, extracellular carboxylesterase B domain spanning amino acids 97 to 681 as well as a transmembrane helix at amino acids 73 to 93 and a cytoplasmic domain at positions 1 to 73, which corroborates that this enzyme is embedded in the cellular membrane as an integral part of it. Gen Ontology Enrichment Analysis at Genontology.org [21], concurs with this assessment and assigns number GO:0016021 to this sequence, predicting that the *F. hepatica* carboxylesterase plays a role as an integral component of the membrane and further suggests that it may be part of the cytochrome bo_3_ ubiquinol oxidase system pathway. The probable function of this carboxylesterase in drug metabolism is independently suggested by KEGG orthology analysis at genome.jp [22,23], assigning KEGG No. K03927, which describes this protein as carboxylesterase type 2 (EC:3.1.1.1 3.1.1.84 3.1.1.56) and suggests a probable role in drug metabolism pathways in conjunction with other enzymes. Both GO and KEGG numbers are compatible with our experimental analysis. Since most carboxylesterase-specific activity has been found to bind to the membrane fraction as a homodimer complex, experimental evidence is compatible with an enzyme working as an integral component of the cellular membrane (Figure 3). The enzymatic activity found in the soluble fraction is consistent with a disrupted monomeric enzyme produced during the preparation of parasite protein extracts.

The mRNA transcript now converted to cDNA was submitted to GenBank, where it was assigned the entry number MT843326, and it can be directly analyzed by pBlast against the nonredundant protein dataset [24,25]. The results are shown in the phylogenetic tree displayed in Figure 4. The 20 highest identities were obtained against parasitic flatworm proteins, and only a handful of these sequences have been correctly described as carboxylesterases or esterase-like proteins, such as carboxylesterase 5A of the cestode *Hymenolepis microstoma*, a carboxylesterase of the trematode *Paragonimus heterotremus*, and three neuroligins, nonenzymatic proteins closely related to cholinesterase, in Echinococcus granulosus, *E. multilucularis* and *Clonorchis sinensis*. The rest of the highest identity proteins were either hypothetical or unnamed proteins of flat worms (Figure 3). The resulting phylogenetic tree obtained by the pBlast toolkit clearly identifies two major clades with a perfect distinction between cestodes and trematodes; notably, the *Fasciola* genus within the trematode clade was set apart in a distinctive subclade of its own, where our carboxylesterase B, GenBank entry MT843326, can be located with 99.7% identity to the same protein reported previously and independently in GenBank, but labeled as hypothetical. Additionally, within the Fasciola genus subclade is the equivalent hypothetical protein in *Fasciola gigantica* with 98% identity. Esterase F4 of the lepidopteron *Galleria mellonela* was included as an out-group for the enhanced accuracy of the phylogenetic analysis (Figure 4).

The bioinformatics analysis of transcript 473 did not find a direct biochemical pathway that may demonstrate an enzymatic effect on benzimidazole-derived anthelmintics; this does not mean that the carboxylesterase B does not interact with secondary metabolites, our study suggests that this enzyme may be part of several metabolic pathways that are implicated in the metabolism of drugs and their metabolites. We already know that triclabendazole anthelmintics have been designed to be metabolically transformed to a more toxic molecule for parasitic helminths [17]; therefore, we are not looking for the catalytic properties of carboxylesterase B on known chemical structures of anthelmintics. Instead, we are looking at the catalytic properties of carboxylesterase B on secondary metabolites of benzimidazole-derived anthelmintics, and our study identified several metabolic pathways that may yield a probable substrate for carboxylesterase B yet to be detected.

## 4. Materials and Methods

### 4.1. Animals and the Parasite Strain

Animal management was performed according to the ethical guidelines of our institutions, and animal use was performed according to the Mexican norm NOM-062-ZOO-1999, and its technical specifications for the production, care and use of laboratory animals. The *F. hepatica* reference strain was maintained as an anthelmintic bioassay reference at the National Center for Disciplinary Research in Veterinary Parasitology (INIFAP, Jiutepec, Morelos, Mexico) and previously registered as NCBI-BioSample SAMN16822856 [3], 90 metacercariae were orally inoculated in a six-months-old rabbit. Parasitic development was confirmed four months later by the presence of liver fluke eggs in the feces. The rabbit was sacrificed, and the liver was removed and dissected. The parasites were recovered from the liver, and a group of twenty parasites was collected in 20 mL of Minimal Essential Medium Eagle (MEM, Sigma Chemicals, St. Louis, MO, USA) supplemented with 10% fetal bovine serum (FBS GIBCO ). 

### 4.2. Enzyme Analysis

For protein extraction purposes, a sample of five parasites was washed in phosphate buffered saline (PBS) at a pH of 7.2, frozen at –196 °C, and macerated into a fine powder according to a previously published procedure [26]. The macerated samples were homogenized (1:1) in PBS at a pH of 7.2, and centrifuged at 5000× *g* for 5 min, and the resulting supernatant was considered crude extract for carboxylesterase-specific activity purposes. The crude extract was centrifuged at 20,000× *g* for 30 min. The resulting pellets were considered the insoluble membrane fraction, and the supernatant was considered the soluble cytosol fraction. All fractions were collected for determination of the protein content using the Lowry method with bovine serum albumin as a standard [27], The protein extracts were stored at −80 °C until assayed. The protein samples were used to determine the carboxylesterase enzymatic activity in *F. hepatica* using α-naphthyl acetate coupled to the diazonium salt Fast Gardner (Sigma Chemicals, MO, USA) as a previously reported substrate [28] on 100 μg of protein extracts from each sample assayed with four replications. The amount of α-naphthol-Azo dye released by the enzymatic reaction was measured spectrophotometrically at 524 nm. Absorbance values were converted to micromoles of hydrolyzed substrate/minute/milligram of protein according to a previously described method [28]. 

SDS-PAGE was performed on 12.5% gels prepared according to a previously reported method [29]. A commercial prestained protein mass standard (Thermo-Fisher, Walthman, MA, USA) was used as a mass reference, and Lab4^®^ algorithm molecular mass calibration was performed. Five micrograms of *F. hepatica* from each protein extract from the parasites was loaded on each well. SDS–PAGE was applied at an electric current of 100 volts for 90 min, and afterward it was processed for carboxyl esterase identification using the method of α-naphthyl acetate coupled to the diazonium salt Fast Gardner as a previously reported substrate [20]. The presence of carboxylesterase bands was recorded in digital images and used to estimate the isozyme molecular masses using the Lab4^®^ algorithm.

### 4.3. High throughput mRNA Sequencing

(1) RNA extraction. Five parasites were frozen at –196 °C and macerated into a fine powder; RNA was extracted using the phenol-chloroform procedure as described previously [29]. RNA quality, integrity, 28S/18S RNA ratio and fragment length distribution, were assessed by capillary electrophoresis using an Agilent 2100 Bioanalyzer. For estimation of the total RNA sample QC such as the RNA concentration and RIN value, a Nanodrop^®^ was used as previously described [3]. 

(2) Transcriptome sequencing was carried out by the BGI Genomics Co. Ltd. Using the Illumina HiSeq 4000 platform. For transcriptome assembly the *F. hepatica* reference genome GCA_900302435.1 was used for transcriptome assembly and differential expression differential display in FPMK. The BLAST similarity analysis was performed using the assembled transcriptome; Gene Ontology was performed according to a previously reported method [21], and KEGG orthology terms associated with the transcripts were extracted from the UniProt, Phyre2 and WormBase-ParaSite databases, integrated with the BLAST search results by the sequencer according to previous reports [22,23,30] and delivered as part of the sequence report. In a comprehensive search for *F. hepatica* esterase DNA sequences within the complete *F. hepatica* genomes reported in Bioprojects PRJEB6687, PRJEB25283 and PRJNA179522 as well as in the complete transcriptome reported in Bioproject PRJNA330752 using WormBase Parasite as well as GenBank search toolkits as previously described [24,25], every esterase found in the databases was searched for similar sequences in our transcriptome by the BLAST similarity analysis according to a previous report [31]. The obtained *F. hepatica* transcriptome database was made public at GenBank BioProject PRJNA649302, BioSample SAMN16822856, RNAseq data SRR13076126 (https://trace.ncbi.nlm.nih.gov/Traces/sra/?run=SRR13076124) and previously described elsewhere [3].

### 4.4. Phylogenetic Analysis

The deduced carboxylesterase B cDNA and amino acid sequence were submitted to the GenBank, assigning the entry number MT843326. The analysis of this sequence by the pBlast algorithm at the NCBI website (blast.ncbi.nlm.nih.gov accessed on 8 November 2021) against the nonredundant protein sequences bank was performed according to a previous report [31]. Those protein sequences with an identity score above 30% were selected for phylogenetic analysis offered by the pBlast Tree View (www.ncbi.nlm.nih.gov/blast/treeview/ accessed on 8 November 2021) by the method of fast minimum evolution [19], generating an unrooted tree with a maximum sequence difference of 0.85.

## 5. Conclusions

Our study suggests that the *F. hepatica* carboxylesterase B is an essential enzyme for the metabolism of the liver fluke; our bioinformatics analysis of this gene showed that the enzyme is highly conserved in parasitic flatworms, suggesting a fundamental role in parasitic helminth physiology. The literature inquiry describes these enzymes as part of the xenobiotic metabolizing group of enzymes, and some of them may be involved in drug metabolism [9] our bioinformatics study confirms this assessment and suggests several metabolic pathways that need further studies. The carboxylesterase studied here was discovered to be induced by the presence of benzamidine-derived anthelmintics in parasitized sheep hosts [17], and our study confirmed its presence and elaborates on the possible role in drug metabolism. Our bioinformatics analysis further suggest that this 85 kDa enzyme works as a 170 kDa dimer complex as an integral component of the cellular membrane interacting with several other enzymes [8], some of them known XMEs. Our study shows that the identified 3146 bp mRNA transcript coding a 2205 bp open reading frame translates into a protein of 735 amino acids that has a putative carboxylesterase B functional enzymatic activity undisclosed until now; the gene coding for this enzyme was found to be highly conserved in other reported complete *F. hepatica* genomes and suggests that this enzyme plays an important role in other flatworms. However, until now, no evidence of the functionality of these putative genes existed, and they were labeled hypothetical proteins. The carboxylesterase B activity can be assayed on synthetic chromogenic substrates, something useful for biochemical testing in follow-up studies. Additionally, our study now makes DNA data for this carboxylesterase available to the scientific community for follow up studies.

## Figures and Tables

**Figure 1 pathogens-10-01454-f001:**
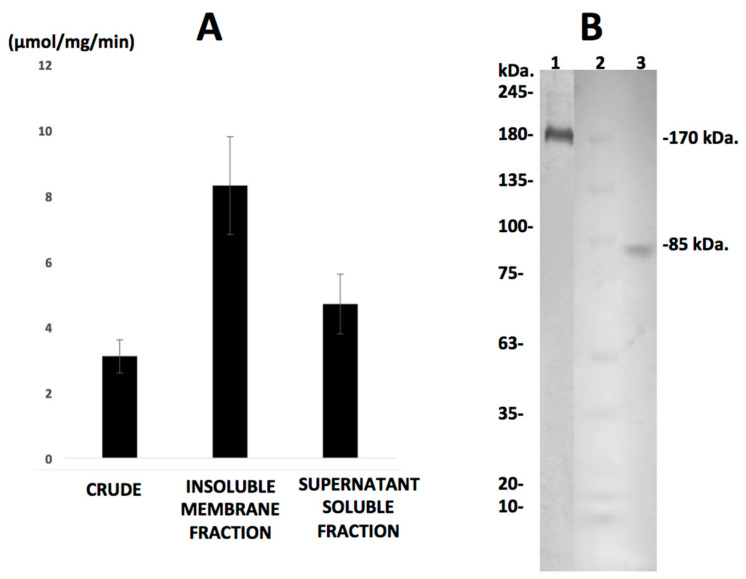
*Fasciola hepatica* Carboxyl esterase specific activity and SDS-PAGE zymograms. (**A**): Enzyme-specific activity from parasite′s protein extracts measured by hydrolysis of the chromogenic synthetic substrate α-naphtyl acetate coupled to the diazonium salt Fast Gardner at 524 nm. (**B**): Non-reducing SDS-PAGE zymograms of the insoluble membrane fraction (lane 1) and the soluble fraction (lane 3), a carboxylesterase enzyme of 85 kDa is detected in the soluble fraction whereas, a 170 kDa enzyme was found in the membrane fraction. Lane 2 contains a commercial pre-stained protein molecular mass reference.

**Figure 2 pathogens-10-01454-f002:**
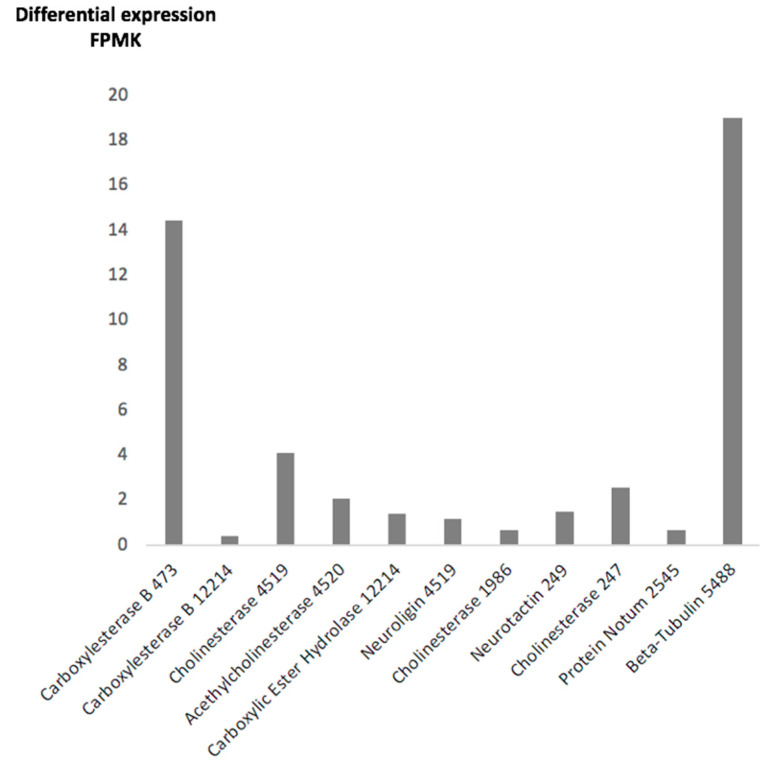
Differential expression of mRNA transcripts with predicted esterase enzymatic specific activity. The differential expression levels in fragments per million kilobases (FPMKs) from every esterase in our transcriptome were determined during the sequencing procedure, annotated in Table 1 and graphically represented. Gene onthology predicted function and transcript number at the SRR13076124 database were included at the footnote for every transcript. The constitutive gen Beta-Tubulin found in our transcriptome as transcript 5488, was used as high expression reference.

**Figure 3 pathogens-10-01454-f003:**
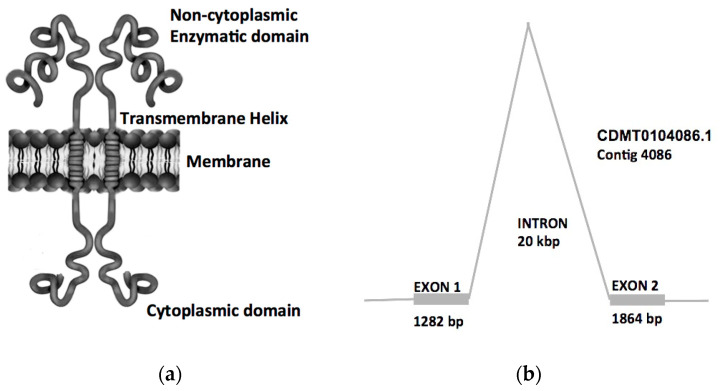
Bioinformatics predicted polypeptide domains (**a**) and exons-introns analysis (**b**). The Interpro algorithm predicted a non-cytoplasmic carboxylesterase domain, a trans-membrane helix domain and a cytoplasmic domain. The transcript shows identity against the GENBANK *Fasciola hepatica* complete genome CDMT01004086.1 contig 4086 showing two exons of 1282 and 1864 bp respectively, and an unusually large intron of 20 kbp.

**Figure 4 pathogens-10-01454-f004:**
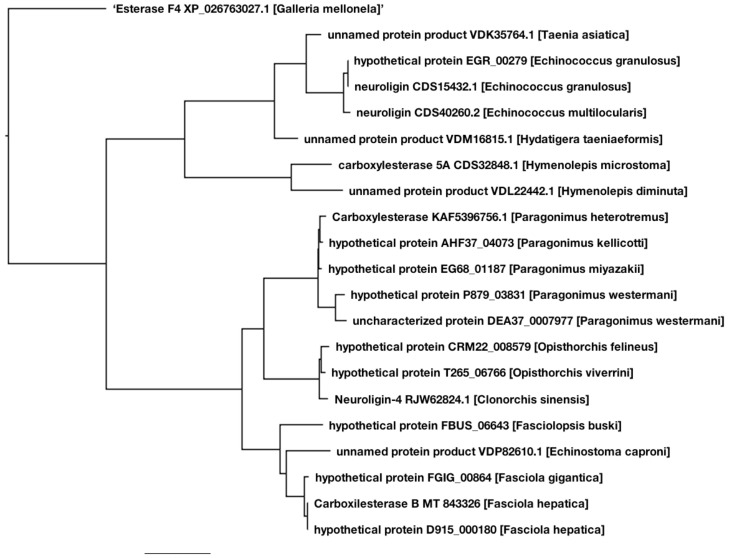
Carboxylesterase B deduced cDNA and amino acid sequence was submitted to the GENBANK assigning the entry number MT843326, the analysis of this sequence by pBlast algorithm at NCBI web site (blast.ncbi.nlm.nih.gov accessed on 8 November 2021) against the non-redundant protein sequences bank. Those protein sequences with an identity score above 30%, were selected for phylogenetic analysis offered by the pBlast Tree View (www.ncbi.nlm.nih.gov/blast/treeview/ accessed on 8 November 2021) by the method of Fast Minimum Evolution [19], generating an un-rooted tree with a maximum sequence difference of 0.85.

**Table 1 pathogens-10-01454-t001:** mRNA transcripts with predicted esterase enzymatic specific activity. The *F. hepatica* transcriptome made public as RNAseq data SRR13076124 available on-line at https://trace.ncbi.nlm.nih.gov/Traces/sra/?run=SRR13076124 (accessed on 8 November 2021), identified ten transcripts labeled esterase by Gene Ontology/Orthology, and two additional sequences tagged as esterase were not represented in our transcriptome but were found in the *F. hepatica* complete genome datasets available at the GenBank and WormBase-ParaSite databases.

Transcript SRR13076124 Spot No.	Size bp	AA	Molecular Mass	Expression FPKM	Description	Genome Identity UNIPROTKB/Description	Genome Identity GenBank/Description
473	3146	735	83.5	14.45	Carboxylesterase B	A0A4E0S0J7/Coesterase	D915-000180/Hypotetical
12214	1837	424	25.41	0.42	Carboxylesterase B	No match	KZ430433.1/ Juvenile hormone esterase
4519	3482	697	78.8	4.11	Cholinesterase	AOAA4E0RPG5/Neuroligin	THD27534.1/Cholinesterase
4520	1816	383	44.0	2.1	Acetylcholinesterase	AOAA4E0RPG/Carboxylic ester hydrolase	THD27534/Cholinesterase
12214	1837	430	48.4	1.4	Carboxylic Ester Hydrolase	A0A4E0S092 /Uncharacterized protein	D915_006009/Hypotetical
4519	2094	697	78.8	1.2	Neuroligin	D915_001711/Carboxylic ester hydrolase	THD27534/Cholinesterase
1986	1424	502	56.9	0.7	Cholinesterase	A0A2H1CBM8 Cholinesterase	PIS84875.1 /Neuroligin
249	2085	694	79.7	1.5	Neurotactin	A0A4E0RJZ9/Neurotactin	THD28606.1/Neurotactin
247	1510	490	56.19	2.6	Cholinesterase	A0A2H1CVT6/Cholinesterase	THD28606/Neurotactin
2545	2304	606	69.2	0.7	Uncharacterized	0A2H1CVW9/Uncharacterized	THD28213.1/Protein Notum
No match	1934	644	72.7	No data	Cholinesterase	A0A2H1CPF2/Carboxylesterase	PIS89339/ Cholinesterase
No match	1689	562	63.1	No data	Carboxylesterase	No match	KZ429703.1/Carboxylesterase

## Data Availability

Access to RNAseq data Transcriptome Adult TCBZ resistant, Tanscriptome dataset is available at https://www.ncbi.nlm.nih.gov/sra/SRX9523046 accessed on 8 November 2021: RNAseq data Transcriptome Adult TCBZ resistant: https://www.ncbi.nlm.nih.gov/sra/SRX9523047[accn] accessed on 8 November 2021., https://www.ncbi.nlm.nih.gov/biosample/16822857 accessed on 8 November 2021. NCBI-BioSample SAMN16822858 (Adult *Fasciola hepatica* Triclabendazole resistant). RNAseq data SRR13076124 available on-line at https://trace.ncbi.nlm.nih.gov/Traces/sra/?run=SRR13076124 accessed on 8 November 2021. and tables that includes differential expression for each transcript in FPKM, GO and KEGG numbers as well as GENBANK accession numbers for each sequence, details of raw read generated, assembly and annotation information, overall transcriptomic annotation information such as mapping rate, number of known and unknown transcript identified, splicing events and long noncoding RNA transcripts as well as the annotated gene ontology divided in number of genes found as cellular components or fulfilling a biological process or molecular function.

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
