# Peer review of "Transcriptome-Based Identification of a Functional Fasciola hepatica Carboxylesterase B"

_pathogens, 2021, doi:10.3390/pathogens10111454_

Round 1

Reviewer 1 Report

1: Ln 25 and 154 and 225 states the protein Mw is 83.5 kDa but it is noted as 83.7 kDa in Table 1. One needs changing for consistency.  

2: Ln 25 “ …a putative carboxylesterase B enzymatic activity” needs rewording to “a putative carboxylesterase B enzyme” 

3: Ln 26 – 27. Maths for 1282 bp + 1864 bp + 20,000 bp does not equal 23,230 bp like stated. 

4: Ln 32. “zymogram on membrane and cytosolic fractions” The extraction for your membrane fraction is more accurately an insoluble fraction containing membrane proteins. It needs to be noted as insoluble membrane fraction here and throughout. Therefore, the statement “both traits were experimentally confirmed by PAGE zymogram” needs to be re-phrased as there is only implications by PAGE that the protein is an integral protein.  

5: Ln 46. “ use of massive amounts of anthelmintics on the livestock” this implies the misuse or overuse of anthelmintics. Can it be rephased so its clear one animal is not receiving a massive amount of drug.  

6: Ln 46 – 48. An additional reference is needed for the sentences idea. Ref 3 does not seem to cover the mechanisms leading to resistance.  

7: Ln 63-69. “role in the resistance of a variety of parasites resistance against antiparasitic compounds” repetitious. Could the word mechanisms be added to improve the statement? 

8: Ln 78 and 81. “However” and “Consequently, it is” appears to be in grey text.  

9: Ln 89. A space is required in “F.hepatica” 

10: Ln 94 and Ln188/190 (and throughout). The use of the word isozymes is surprising since the authors are proposing that the two bands at 85 at 170 kDa are homodimers.  

11: Ln 99 and Ln 320. Carboxylesterase does not need a space.  

12: Figure 1B. The zymogram image quality is poor. Check the 60 and 120 marker correctly align. The 170 marker is indistinguishable. Can the author provide an uncropped version or increase the height of the gel as it is hard to tell if the 170 kDa band has entered the resolving gel. Ie aggregation or dimer.  

13: Species name needs to be in italics on Ln 107, 113, 166, 170, 171, 181, 228, 232, 242, 261, 262, 263, 267, 271, 272, 332, 339, 345  

14: Ln 118. “Table 2” I believe it should be Table 1. 

15: The data shown in Figure 2 is clearly displayed in Table 1 what is the justification of repeating the data? The beta tubulin expression level can be noted in the main text along side Table 1. A multiple sequence alignment/ tree of the Carboxylesterase proteins would be a more appropriate/informative secound figure.  

16: Ln 143 and Ln 238. Could the authors check that the noncytoplasmic domain is from “197-681” and not 97-681. Otherwise, there is a missing 100 amino acids.  

17: Ln 151-153. Have the authors consisdered using phyre for dimer prediction to supplement the InterPro information? An alignment to a dimer carboxylesterase and citation of the crytal structure would justify one of the major findings a lot better. Following on: Ln 195. Is there a primary reference looking at monomers and dimers of Caboxylesterases? Ref 21 is a weak citation here.  

18: Ln 232-234. The GenBank entry numbers are not in Table 1. No DNA spacing is represented in Table 1.  

19: Ln 252. Is Figure 2 the correct Figure here.  

20: Ln 297. Heading does not represent the scope of the paragraph. Both 4.2 and 4.3 are enzyme analysis.  

21: Could the authors justify the use of PBS as the extraction buffer and not a more commonly used RIPA (detergents) or Tris buffer for isolation between cytosol and membrane fractions. This extraction technique is unlikely to isolate a true membrane fraction.  

22: Ln 305. “The” no need for capitalisation.  

23: Ln 309, 321, 101. Can a company be given for diazonium salt Fast Gardner. Or is it Fast Garnet like the citation given [19] indicates?  

24: Is the zymogram reducing or non-reducing? Could a more detailed method be added for the zymogram. Have the authors considered any specific inhibitors to ensure the correct activity? There are other experimental methods to determine monomeric and dimeriation. Have the authors considered recombinant protein expression and size exclusion? Or a Coomassie stained reducing and non-reducing SDS-PAGE or native PAGE of both fractions. More experimental evidence for a dimer vs aggregation. As mentioned above is the 170 kDa band sitting at the resolving/stacking gel interface indicating aggregation? Minimal evidence of a homodimer, only an active complex.  

25: Ln 359. “he” should be “the”.  

26: Ln 376. A period is required at the end of the conclusion.  

Author Response

1: Data has been corrected to 83.5

2:  Phrase was reworded as requested

3:  The intron correct size of 20084 was included in the text

4:  The paragraph was rephrased as suggested

5: The paragraph was rephrased as suggested

6: A new reference was added and the paragraph rephrased

7: The paragraph was rephrased as suggested

8: The color of the indicated text was corrected

9: A space was added

10: All isozymes were eliminated

11:spaces were eliminated

12: A better image was uploaded to the manuscript

13: All indicated italics were added

14: Tables numbers were verified

15:We believe that a graphical representation of the differential expression is important for understanding the carboxylesterase identification criteria , the phylogenetic tree displayed in figure 4 is based in a multiple sequence analysis and the alignment itself would take a big space so we opted for showing only the tree.

16: The range on non-cytoplasmic domain was corrected

17: We did use Phyre2 and added a citation to the text, an additional citation was added Ln 195.

18:Table 1 only describe the properties of the transcripts after gen ontology enrichment analysis, Ln 132-234 refer to Blast identities to similar sequences in the GENBANK against carboxylesterase B.

19:Figure number was corrected

20:Heading 4.3 was eliminated

21 Our only justification is that we did not have access to RIPA, maybe we did not obtain a true membrane fraction but had a close enough result to achieve the experimental evidence we were looking for. 

22: Capitalisation was removed

23: The Fast Garnet company was indicated in M&M

24: A new zymogram image was added to figure 1 to address the stack gel origin visibility, an amendment was made to the text to indicate non-reducing conditions, inhibitors are considered for a follow up study

25:Modified as suggested 

26: Period added

27: I appreciate your review it was most helpful and constructive something rare in reviewer nowadays.

Reviewer 2 Report

Dear authors, 

The manuscript “Transcriptome-based identification of a functional Fasciola hepatica Carboxylesterase B” is well written and nicely written with lot of supporting evidence.

Authors identified several putative carboxylesterase transcripts including one with putative carboxylesterase B enzymatic activity.  Authors performed enzymatic activity on F. hepatica protein extracts by SDS-PAGE zymograms using synthetic chromogenic substrates and Bioinformatic analysis for further confirmation.  Their analysis indicated that DNA sequences that code for this particular enzyme are highly conserved in other parasitic trematodes, although they are labeled hypothetical proteins.

There remain some minor issues that authors should consider the comments useful for further revision of the manuscript. Importantly, there are some grammatical errors throughout the manuscript and so it has to read by a native speaker for clarity. It is very important to fix all the grammatical errors before resubmitting the manuscript.

Comments:

  1. 1 A. It would be better to make a bar chart for the numbers provided. It seems it is a table, but not a figure
  2. In Figure 2, authors showed differential expression of mRNA transcripts with predicted esterase enzymatic specific activity. It would be interesting to see if there is a match between gene expression and protein expression. For that purpose, it is important to run western blot in this experiment. In addition it is very important to perform a qRTPCR to validate the RNA seq data.
  3. Lines 131- 134: The following sentence is not clear. Please write them in simple sentences so that readers can understand them very well.

Once all transcripts as well as the DNA esterase sequences not represented in our 131 transcriptome were analyzed, we found that transcript 473 exhibited a level of expres-132 sion similar to a constitutive gene (14.45 FPMKs) as well as the mRNA size required to 133 translate into an 85 kDa protein or a homodimer complex of 170 kDa proteins deter-134 mined by SDS–PAGE zymograms

  1. What are the possible role of these carboxy esterase enzymes? It is important to discuss their roles in context of life style of other parasites.

Author Response

Reviewer 2

    1.    A chart was added to figure 1.

    2.    A qPCR experiment is an ongoing follow up study., gene expression and protein expression correlation was assessed by enzymatic specific activity and zymograms.

    3.    Lines 131- 134: were rephrased

    4.    Possible roles of carboxylesterases in F. hepatica are described

Reviewer 3 Report

  1. Authors should italicize all genus and species names throughout the manuscript.
  2. Line 22-25: This sentence is too long and may need to be broken 
  3. Line 107: Correct spelling to "transcript"
  4. Lines 73-75: Please provide references
  5. Line 75: “During this study ….. please clarify which study is been referred to here.  
  6. Lines 277-283: The state seems confusing. Please rephrase to provide clarity.
  7. Lines 364-365: How did the author know they were dealing with the same enzyme?
  8. The conclusion needs to be rephrased i.e. tailored specifically to the experiment carried out.
  9. Line 359: ‘he” should be corrected to “the”
  10. Lines 362-364: Sentence is not necessary for a conclusion, This same statement has been presented in the discussion. I suggest it be merged with lines 359-360 to read as follows; “Our study …………….. metabolism of liver fluke, probably drug metabolism.

Author Response

    1.    Modification were made as requested

    2.   Sentence modified as suggested

    3.   Spelling was corrected

    4.    Reference was provided

    5.    Reference was provided

    6.    Lines 277-283: were rephrased

    7.    Lines 364-365: these enzymes had an identity above 80% in different flatworm species they can be considered analogous enzymes

    8.    Conclusión was rephrased

    9.    Line 359: Corrected as suggested

    10.      Conclusión was rephrased